# Cytoreductive Surgery (CRS) and HIPEC for Advanced Ovarian Cancer with Peritoneal Metastases: Italian PSM Oncoteam Evidence and Study Purposes

**DOI:** 10.3390/cancers14236010

**Published:** 2022-12-06

**Authors:** Daniele Marrelli, Luca Ansaloni, Orietta Federici, Salvatore Asero, Ludovico Carbone, Luigi Marano, Gianluca Baiocchi, Marco Vaira, Federico Coccolini, Andrea Di Giorgio, Massimo Framarini, Roberta Gelmini, Carmen Palopoli, Fabio Accarpio, Anna Fagotti

**Affiliations:** 1Unit of General Surgery and Surgical Oncology, Department of Medicine, Surgery and Neurosciences, University of Siena, 53100 Siena, Italy; 2Unit of General Surgery San Matteo Hospital, 27100 Pavia, Italy; 3Surgical Oncology, Peritoneum and Abdomen Pathologies, National Cancer Institute Regina Elena, 00144 Rome, Italy; 4Unit of Surgical Oncology, Soft Tissue Tumors, Department of Oncology, Azienda Ospedaliera di Rilievo Nazionale e di Alta Specializzazione Garibaldi, 95123 Catania, Italy; 5Department of Clinical and Experimental Sciences, University of Brescia, ASST Spedali Civili, 25123 Brescia, Italy; 6Candiolo Cancer Institute, FPO–IRCCS, Candiolo, 10060 Torino, Italy; 7General, Emergency and Trauma Surgery, Pisa University Hospital, 56122 Pisa, Italy; 8Surgical Unit of Peritoneum and Retroperitoneum, Fondazione Policlinico Universitario A. Gemelli–IRCCS, 00168 Rome, Italy; 9Surgery and Advanced Oncological Therapy Unit, Ospedale GB. Morgagni-L. Pierantoni, AUSL Forlì, 47121 Forlì-Cesena, Italy; 10Unit of Emergency General Surgery and Surgical Oncology, AOU Policlinico di Modena, 41125 Modena, Italy; 11Unit of PSG and OBI, Azienda Ospedaliera Universitaria G. Martino, 98124 Messina, Italy; 12CRS and HIPEC Unit, Pietro Valdoni, Umberto I Policlinico di Roma, 00161 Roma, Italy; 13Unit of Ovarian Carcinoma, Fondazione Policlinico Universitario A. Gemelli–IRCCS, 00168 Rome, Italy

**Keywords:** advanced ovarian cancer, peritoneal metastases, neoadjuvant chemotherapy, hyperthermic intraperitoneal chemotherapy, cytoreductive surgery

## Abstract

**Simple Summary:**

Ovarian cancer is still the most lethal gynecologic malignancy with a median 5-year survival of less than 50%. Most cases are diagnosed at the advanced stages and only limited improvements in overall survival are achieved by standard treatments. Despite the paucity of randomized clinical trials, combined cytoreductive surgery and hyperthermic intraperitoneal chemotherapy (HIPEC) show encouraging survival outcomes in primary ovarian cancer with peritoneal metastases. It gives rise to the definite need to redefine not only the best treatment option, but also the optimal treatment sequence based on the primary tumor and the extent of disease.

**Abstract:**

Ovarian cancer is the eighth most common neoplasm in women with a high mortality rate mainly due to a marked propensity for peritoneal spread directly at diagnosis, as well as tumor recurrence after radical surgical treatment. Treatments for peritoneal metastases have to be designed from a patient’s perspective and focus on meaningful measures of benefit. Hyperthermic intraperitoneal chemotherapy (HIPEC), a strategy combining maximal cytoreductive surgery with regional chemotherapy, has been proposed to treat advanced ovarian cancer. Preliminary results to date have shown promising results, with improved survival outcomes and tumor regression. As knowledge about the disease process increases, practice guidelines will continue to evolve. In this review, we have reported a broad overview of advanced ovarian cancer management, and an update of the current evidence. The future perspectives of the Italian Society of Surgical Oncology (SICO) are discussed conclusively.

## 1. Introduction

Ovarian cancer (OC) affects 239,000 women and causes 152,000 deaths every year [1,2]. Overall, it ranks 8th among cancer in women (3.6% of all cancer cases) and 8th among causes of female cancer-related death (4.3% of all cancer deaths) [3,4]. What’s more, the rate of OC is steadily increasing, and recent data indicate a mortality-to-incidence ratio of 0.6 [5,6], reflecting a dramatic diagnostic delay [7]. Although early stage OC is highly curable, 4/5 women present with advanced disease [8,9,10], after the cancer has leaked from the pelvic cavity [11,12].

The International Federation of Gynecology and Obstetrics tumor staging system (FIGO 2018) proved to have higher prognostic power than the American Joint Committee on Cancer (AJCC) system, both based on the tumor–node–metastasis (TNM) parameters [13,14,15]. Peritoneal metastases are staged as FIGO III, which includes disease that has spread from the ovaries with microscopic peritoneal involvement (IIIA2), visible peritoneal implants outside the pelvis (IIIB) and greater peritoneal metastasis (IIIC), with or without retroperitoneal lymph node involvement. FIGO IIIA2, IIB and IIIC represent more than 60% of the cases of OC [11,16,17]. (Table 1) In 2015, the 10-year overall survival (OS) for patients diagnosed with early stage (I–II) OC was 55% [6,9], versus 15% for those with advanced-stage disease (III–IV) [18,19]. Nevertheless, median progression-free survival (PFS) has increased from 12.7 months in 2004 to 20.7 months in 2016 in FIGO III-IV. An increased centralization in high-volume centers has facilitated the enrolment of patients in ongoing trials, allowing to profile distinct clinicopathological and molecular features and to purpose a breakthrough tailored medicine [20]. 

OC originates from the epithelial cells, located on the ovarian surface or distal fallopian tube epithelium in about 90% of cases, with the histological subtype high-grade serous accounting for 70% of cases [22]. A dualistic model was proposed to classify epithelial OC into two groups [23], type I and type II, that progress along two different tumorigenic pathways [24]. The type II group is mainly composed of high-grade serous carcinomas, which develop from intraepithelial carcinomas in the fallopian tube and disseminate as carcinomas that involve the ovary and extraovarian sites. Despite the criticism that type I encompasses a wide variety of tumor subtypes, this model firstly described high-grade serous and low-grade serous OC as two distinct entities [25]. Moreover, it is important to note that type I tumors diagnosed at an advanced stage, though rare, have a poor prognosis such as type II tumors [26]. 

Nowadays, OC is treated as a single disease [10,27], and tumor heterogeneity (across subtypes, within a single tumor [28], or between a primary tumor and metastasis) represents a major cause of treatment failure [29,30]. This has several implications for our understanding of the disease on a clinical level. The classic linear clonal evolution model, consisting of an addition of mutations that will lead to the development of the metastasis, has been replaced by an evolutionary tree. Single cancer cells evolve independently from the primary clone, gaining the function of metastasis-to-metastasis spread. Thus, a single biopsy is not representative of the entire tumor, and metastases could harbor targetable mutations which are absent in the primary tumor [31,32].

All qualities (tumor stage, tumor subtype and quantitatively measuring tumor heterogeneity) are essential principles that guide the treatment decision-making process and shape the direction of clinical research [33]. Moreover, molecular and genetic features, including germline and somatic breast cancer 1 (BRCA1) and breast cancer 2 (BRCA2) mutations [34,35], may influence the prognosis of patients [36,37], the recurrence rate [38], and the cancer response to DNA-damaging agents, such as platinum chemotherapy [39], or DNA repair pathways inhibitors. Poly-ADP-ribose polymerase inhibitors (PARPi) have recently shown to be effective as a first-line or maintenance treatment for patients with advanced BRCA-mutated ovarian cancer [40]. Other targeted therapies, immune checkpoint inhibitors and antiangiogenic agents could have a synergistic effect, as abnormal tumor vasculature may prevent immune cell trafficking [41].

OC management is best provided by an expert multidisciplinary team, including radiologist, pathologist, gynecological surgeon and medical oncologist [1,42,43,44], who evaluate and re-elaborate differentiated therapeutic opportunities according to clinical information, pursuing a circular approach with the patient in the middle [15]. As described by Mercado et al. [45], patients treated in referral centers have a 40% higher OS compared with patients treated in a peripheral center. The management of OC is evolving from a “one size fits all” approach to “personalized” treatment, in which surgeons are becoming more precise about using surgery, chemotherapy, molecular targets and evolving drugs [7].

In this review, we provide an overview of the Oncoteam’s experience, evidence and ongoing trials led by the Peritoneal Surface Malignancy (PSM) team of the Italian Society of Surgical Oncology (SICO).

## 2. Treatment of Ovarian Cancer Metastatic Peritoneal Surface Malignancies

PSM were considered terminal diseases with limited therapeutic options and heralded a poor prognosis [46,47]. A step change occurred at the end of the last millennium [48], as a result of the adoption of novel surgical techniques, such as peritonectomy procedures and multivisceral resections (1995) [49], and the application of intraperitoneal chemotherapy (1980) [50,51,52,53,54]. Aggressive peritoneal therapies are based on the premise that cancer isolated to the peritoneal cavity is a locoregional disease, exhibiting unique behaviors compared with extraperitoneal metastatic disease [55]. Regarding OC, treatment decisions are mainly based on the disease stage and biology, prior therapy and comorbidities.

### 2.1. CRS in Primary Ovarian Cancer with Peritoneal Metastases

The standard first-line treatment of advanced OC consists of cytoreductive surgery (CRS), to remove as much of the cancer as possible, followed by taxane and/or platinum-based chemotherapy (every three weeks for six cycles) [1,3,56]. More than two chemotherapy drugs in combination do not improve outcomes. CRS or debulking surgery is defined as “primary” if it is aimed at completing the resection of all macroscopic tumors in patients with their first diagnosis of advanced OC before any other treatment; “interval” when following neoadjuvant chemotherapy (three to six cycles additional cycles) [16,57]; and “consolidation” after a complete response to neoadjuvant chemotherapy [58].

Nowadays, preoperative chemotherapy is administered in high-volume diseases (i) to improve tumor resectability [59,60], (ii) to perform a test of in-vivo sensitivity selecting good responders to hyperthermic intraperitoneal chemotherapy (HIPEC), (iii) in women with a poor performance status at presentation due to high-risk comorbidities [61,62] or (iv) to impede the growth of resistant cancer cells that remain after the first round of chemotherapy, delaying disease progression [63,64,65,66,67,68].Rosen et al. reported that 50.1% of patients treated with neoadjuvant therapy achieved a status of no residual disease, compared with 41.5% of patients who underwent primary debulking surgery [69]. Surprisingly, 7-year OS rates were strongly different between interval and primary treatment groups (8.6% vs. 41%, respectively), suggesting the role of chemotherapy in camouflaging microscopic tumor foci remaining after surgery. After the European Organization for Research and Treatment of Cancer (EORTC) trial firstly demonstrated the noninferiority of neoadjuvant therapy followed by interval debulking surgery, it was considered in a wider breadth of patients [70,71]. However, while the CHORUS trial showed similar results [72], the MSKCC trial showed prolonged survival when compared with patients in EORTC study [73]. On the other hand, JCOG0602 failed to demonstrate non-inferiority [74] and SCORPION failed to demonstrate the superiority of preoperative therapy [75]. 

The next SUNNY trial will evaluate the efficacy and safety of interval CRS in AJCC TNM stage IIIC or IV epithelial OC [76]. The next TRUST trial will compare primary CRS followed by six cycles of platinum-based chemotherapy vs. interval CRS (three cycles of neoadjuvant chemotherapy and three cycles of platinum-based post-operative chemotherapy), in order to clarify the optimal timing of surgical therapy [77]. Finally, the next CHRONO trial will assess the impact of interval CRS after three cycles of neoadjuvant chemotherapy (paclitaxel and carboplatin) compared with delayed surgery after six cycles of neoadjuvant chemotherapy on PFS, in FIGO IIIB–IVA patients unsuitable for complete primary surgery [78]. 

In addition, host factors, such as BRCA1 and BRCA2 status, predict the response to chemotherapy and, therefore, the time to recurrence, but do not affect long-term OS [5,60,79,80,81].

As reported in the literature, CRS is considered “optimal” if the residual tumor is less than 1 cm in maximum diameter or thickness [82]. However, the concept of “optimal debulking” is usually dangerously confused with the complete tumor resection and no residual disease (R0) among the non-expert community [19]. Complete CRS, according to the Gynecologic Cancer Inter Group (GCIG), is a state of no visible residual disease, associated with a significantly increased OS and PFS [71,83,84]. A large meta-analysis recently concluded that a residual tumor is a more powerful prognostic determinant than the FIGO stage [19,85,86]. Biological, genomic and molecular features have been investigated to develop predictive signatures that are correlated with optimal and R0 debulking [87,88,89]. On the other hand, a standardized lymphadenectomy has not yet been designed [90], while currently, the removal of bulky lymph nodes is carried out as part of an attempt to achieve maximum cytoreduction [91]. Moreover, despite a trend toward a longer OS in the total parietal peritonectomy, which comprised removal of the entire parietal peritoneum and the greater and lesser omental rather than selective parietal peritonectomy, its use is more liberally applied [92].

Currently, the consensus standard for medical therapy is a combination of carboplatin and paclitaxel, both administered every 3 weeks. Dose-dense chemotherapy should not be considered as a standard because of the lack of evidence in Western studies [93,94,95]. 

Adjuvant chemotherapy is recommended to begin as soon as possible, possibly after 2–4 weeks, as longer delays lead to worse outcomes [96,97,98].

Despite excellent treatment responses in around 70% of women, most patients develop recurrence within the next 3 years [99]. In these cases, second-line or maintenance therapy was performed, given to delay the disease progression [100].

There is much debate about the optimum timing of treatment (neoadjuvant or adjuvant chemotherapy, primary or interval CRS) and the best route of administration (intravenous (IV) or intraperitoneal (IP) therapy). Narod [101], focusing on 20% of women surviving for 10 years or more [102], proposed a holistic model in disease insight: if no residual cancer cells persist in the abdomen after treatment, recurrence is impossible; if no intra-abdominal recurrence develops, the patient is cured [103]. Although there is no complete consensus [104,105], R0 seems to be the highest after primary CRS and IP chemotherapy, while the lowest for women who receive neoadjuvant chemotherapy:Among OC patients who achieve a status of no residual disease through neoadjuvant chemotherapy and interval CRS, an estimated 10% have no residual cancer cells [71,72];If an initial visible residual disease after primary CRS goes to complete tumor resection after adjuvant IV or IP chemotherapy, 18% of patients have R0 [106];If complete tumor resection was obtained after primary CRS, the probability of having no residual cancer cells after adjuvant chemotherapy is estimated to be 33% for those who receive IV chemotherapy and 50% for those who receive IP chemotherapy.

### 2.2. HIPEC in Primary Ovarian Cancer with Peritoneal Metastases

The poor prognosis after standard chemotherapy and the attitude of that remaining, confined to the abdomen, has produced an increased interest in the use of IP therapy. IP chemotherapy takes advantage of the blood–peritoneal barrier to achieve a much higher drug concentration at the peritoneal surface [107,108,109,110]. The addition of hyperthermia to IP chemotherapy implies the thermal enhancement of the chemotherapeutic agents used, increased drug uptake in malignant cells secondary to increased membrane permeability, inhibits repair mechanisms, facilitates lysosomal enzyme activation and selectively improves vascular flow in normal cells [18,111,112].

The first report of the application of HIPEC in OC was in 1994. Since that time, several studies, enrolling large cohorts of women, evaluate the conjunction of HIPEC with CRS [20,113,114], both in the primary and recurrent disease [115,116]. The most recent meta-analysis [116,117,118,119] reported improved PFS after CRS and HIPEC in patients with primary advanced OC, even after adjustment for potential confounders (such as age, stage, neoadjuvant chemotherapy, grade, ECOG status [120], and histology). However, due to the retrospective nature, their results should be interpreted with caution, conditioned by a wide variety of chemotherapeutic regimens [121]. 

The goal of CRS–HIPEC is to remove all visible macroscopic disease while preserving organ function as far as possible. When combined, CRS and HIPEC showed better PFS and OS both in an upfront approach [122], and after neoadjuvant chemotherapy (CARCINOHIPEC trial) [123]. To reach the maximum effect of HIPEC therapy, it should be given when an optimal debulking is performed [19,106,119,124,125,126], achieving a long-term survival rate of around 50%. A multi-center French study compared the consolidation, interval and primary HIPEC for advanced OC, registering an increased OS between groups (33.4 vs. 36.5 vs. 52.7 months, respectively). Moreover, patients performing a well-executed CRS at less than 2.5 mm had a median OS of 41.5 months, compared to 21.2 months for all the others [124]. Patients with residual disease undergoing HIPEC after CRS might increase their life expectancy by 1 year, but do not enhance their chance of a cure [96].

A surgical procedure for OC has been widely described [113,127]. Firstly, the intraoperative re-staging of the disease is performed, and the peritoneal cancer index (PCI) is calculated, based on the distribution and implant size of the metastases as described by Jacquet and Sugarbaker in 1996. Extensive involvement of the small bowel or hepatic hilum is the main contraindication to CRS and HIPEC, as well as the assessment of incomplete resectability with a large residual tumor.

The standard CRS includes the pelvic peritonectomy with en-bloc extra-peritoneal hysterectomy and adnexectomy, until the Douglas pouch (Figure 1). 

The rectum is also en-bloc removed if it is extensively involved by the tumor and a protective ileostomy is performed, whereas small tumor nodules (if present) are removed with rectal preservation. While appendectomy, radical omentectomy and para-aortic and/or pelvic lymphadenectomy (in cases with clinically involved nodes) generally complete the standard procedure, more extensive peritonectomy or visceral resections are less predictable [129], depending on the presence of extra-pelvic peritoneal nodules (Figure 2).

In some selected cases, mesenteric peritonectomy may be indicated, when this is associated with a chance of R0 resection and small bowel wall is not involved (Figure 3).

After CRS, five silicon drains are placed in the abdomen, and temperature probes are positioned above the mesocolon and in the pelvis. Finally, the abdominal wall is closed. In SICO centers, HIPEC is generally performed using a specific device outfitted with two pumps (inflow and outflow), a thermal exchanger and a closed circuit with five liters of perfusate (saline solution). A BSA Mosteller formula calculator estimates the maximum volume with which to fill the abdomen. Finally, after reaching an intra-abdominal temperature of 41–42 °C, chemotherapy agents are injected and circulated with a flow rate of 700–800 mL/min for 60 min [124,130]. Preliminary studies are proposing short-course HIPEC after interval CRS [131], Similarly, Batista et al. proposed an “all-in-one approach”, consisting of neoadjuvant chemotherapy followed by fast-track CRS and short-course HIPEC with cisplatin for 30 min [132].

Among chemotherapy drugs, cisplatin improves OS but is associated with significant toxicity at high dose [133]. While carboplatin is as effective as cisplatin when administered intravenously, the iPocc trial described an improvement in PFS with IP carboplatin plus paclitaxel compared with IV chemotherapy, but this benefit does not translate to OS [134]. A single-center described placitaxel-based HIPEC as safe, given the low rate of morbidity, in neoadjuvant OC [135]. Other studies described no differences in prognosis between cytotoxic and cytostatic agents [136]. An Italian meta-analysis, aimed at focusing on the optimal chemotherapeutic regimen, identified the superiority of the combination of platinum and taxane on PFS in advanced OC at almost all stages of the disease [137]. Moreover, no advantage to an IP therapy trial was registered when bevacizumab was incorporated [138]. Ongoing TRiPocc study will include the molecular biomarker selection of patients for IP chemotherapy. 

To date, two recent phase III randomized controlled trials investigated the role of HIPEC in the treatment of primary OC [3,7,139]: (i) Van Driel et al. described HIPEC following neoadjuvant chemotherapy and interval CRS for newly diagnosed FIGO III epithelial OC (OVHIPEC−1) [140], and (ii) Lim et al. evaluated HIPEC following optimal CRS for FIGO III-IV epithelial OC [141] (Table 2).

The Dutch trial included patients who had a partial or complete response following neoadjuvant chemotherapy and demonstrated a survival advantage (OS 45.7 vs. 33.9 months, *p*-value 0.02; PFS 14.2 vs. 10.7 months, *p*-value 0.003) with the addition of HIPEC (100 mg/m^2^ of cisplatin for 90 min via the open technique at a temperature of 40 °C); 90% of patients completed a full six cycles of chemotherapy. In terms of toxicity, the two groups showed similar results. However, this study was not stratified about important prognostic factors such as BRCA status, FIGO stage or the histologic type of tumor.The Korean trial showed no significant differences in OS and PFS between the HIPEC (75 mg/m^2^ of cisplatin for 90 min via the closed technique at a temperature of 41.5 °C) and no HIPEC arms (69.5 vs. 61.3 months, *p*-value 0.43; 19.8 vs. 18.8 months, *p*-value 0.52). Intra- and post-operative outcomes, such as the extent of surgery, estimated blood loss, residual tumor and length of hospital stay, were not different between both groups. The addition of HIPEC to CRS did not improve survival among patients undergoing primary CRS (*p*-value 0.29; *p*-value 0.51, respectively). Interestingly, the neoadjuvant chemotherapy subgroup showed a trend of improved survival in favor of HIPEC (61.8 vs. 48.2, *p*-value 0.4; 17.4 vs. 15.4 months, *p*-value 0.04).

In 2021, Marrelli et al. [142] proposed a novel treatment protocol with six cycles of neoadjuvant chemotherapy, consisting of paclitaxel plus cisplatin or carboplatin, repeated every 21 days, in women with FIGO stage III OC. According to RECIST criteria [143], the clinical response to neoadjuvant chemotherapy was observed in 83.9% of cases; HIPEC was completed in 85.2%, while adjuvant therapy in 70.4%. The median survival rates in the entire cohort and HIPEC group were 36 and 53 months, while 5-year OS was 36% and 42%, respectively. However, it is unclear if the six cycles approach is associated with better downstaging of the disease than three cycles of neoadjuvant therapy.

To the best of our knowledge, the prognostic effect of HIPEC remains unclear in candidates for primary CRS. OVHIPEC-2 will aim to compare the power of HIPEC (with cisplatinum 100 mg/m^2^) added to primary CRS (no residual disease or up to 2.5 mm in maximum dimension) on OS in FIGO III OC, with acceptable morbidity, when followed by standard adjuvant chemotherapy [144]. Similar endpoints are present in the ongoing EHTASEOCCS trial, which will focus on PFS as the primary outcome [145].

Looking at observational studies [146], a large Spanish case series showed higher PFS in patients treated with primary compared to interval CRS and HIPEC (3-years PFS 63% vs. 18%, *p*-value < 0.01) [147]. An Italian prospective cohort study, enrolling FIGO IIIC and IV women, observed similar intra- and early post-operative complications, as well as the length of stay and the time to start adjuvant chemotherapy between HIPEC and non-HIPEC groups [82].

Interestingly, OC with increased BRCA1 expression has a 36-month survival improvement when treated with IP chemotherapy [148].

Nevertheless, too much bias (different drugs and doses, different perfusion times at different temperatures, different time administration and different criteria for evaluating the completeness of cytoreduction) precludes definitive conclusions about the survival results [55]. Thus, the GCIG in 2016 sets benchmark standards to be adopted for clinical trial design [64,149].

In 2021, the OVHIPEC group, using the EORTC questionnaires, concluded that the addition of HIPEC to interval CRS does not negatively impact health-related quality of life [144]. Opting for therapeutic choices considering the patients’ clinical conditions [150], comorbidities, age and frailty status is necessary to improve their quality of life and life expectancy [151].

Macrì et al. recorded the ECOG score and severe morbidity as predictors of 60-days survival [152,153].

Furthermore, Boerner et al. recommend special consideration for OC patients at higher risk for toxicity: a preoperative creatinine level of 1.0–1.5 mg/dL, non-insulin dependent diabetes mellitus, baseline neuropathy and baseline hearing loss [121].

On the other hand, IP chemotherapy is associated with long operative times, prolonged hospital stays, high morbidity rates [53,154,155,156,157,158,159] and, ultimately, increased hospital costs. While the European Society for Medical Oncology (ESMO) and European Society of Gynecological Oncology (ESGO) failed to recommend HIPEC as the first-line standard of care for OC [1,160], the latest National Comprehensive Cancer Network (NCCN) guidelines encouraged HIPEC at the time of interval CRS in stage III disease, followed by adjuvant IV chemotherapy [56,161].

Although debate is still open, patients undergoing HIPEC showed better (1) survival outcomes, (2) tumor regression and (3) quality of life. “One coincidence is just a coincidence, two coincidences are a clue, three coincidences are a proof.” Believing in Poirot’s philosophy, more research on this technology seems warranted and might be considered in the fight against OC [162]. SICO has been pursuing these goals for over 10 years.

## 3. 1st Evidence−Based Italian Consensus Conference on CRS and HIPEC for Peritoneal Metastases from Ovarian Cancer

In 2017, a multidisciplinary Italian research group attempted to trace a national consensus unanimously [163] agreed regarding the treatment of PC of ovarian origin. It brought together the Italian Society of Surgical Oncology (SICO), the Italian Society of Obstretics and Gynaecology (SIGO), the Italian Association of Hospital Obstetricians and Gynaecologists (AOGOI) and the Italian Association of Medical Oncology (AIOM). An expert committee prepared questions on the diagnosis and staging work-up, indications and techniques of peritonectomy, protocols of systemic chemotherapy as well as HIPEC. These were presented and discussed among the jury panel of experts on the basis of the most updated scientific literature, assigning a definite statement to each topic tailored to the Italian context (Figure 4a). 

The Italian evidence-based consensus clearly stated the “high level” recommendations regarding the peritonectomy procedure, such as the standard report of CC score at the end of the surgery [19,164,165,166], the CC0 as the desirable surgical outcome of CRS and the same type of surgery in primary as well as after neoadjuvant settings. However, the locoregional selective peritonectomy in case of carcinosis limited to certain quadrants has reached a recommendation of “low” grade. Similarly, selective peritonectomy in case of suspected residual cancer after chemotherapy as well as regions involved in the diagnosis but that had responded to chemotherapy are still uncertain. Anyway, the interesting question moved to expert panelists was if HIPEC has a role in the treatment of PSM from OC. Data from the analyzed literature highlighted HIPEC as a treatment associated with better survival rates than chemotherapy alone [167]. In the same way, comparable survival outcomes were registered between patients treated with HIPEC and patients submitted to secondary surgery followed by chemotherapy [167]. Therefore, even when with a low level of evidence, HIPEC is recommended alongside secondary surgery in platinum-sensitive relapses. 

The completeness of surgery is a relevant prognostic factor of FIGO III-IV patients treated with platinum-based chemotherapy [168,169,170,171,172,173]. Interestingly, the Italian consensus conference did not consider the ethical or advisable recommendation of CRS+HIPEC in patients with a CC score greater than 1 for advanced OC as well as for peritoneal relapses, with a low level of evidence.

## 4. Timing of CRS and HIPEC in Primary Ovarian Cancer with Peritoneal Metastases

Combined HIPEC with peritonectomy represents the standard of care in Italian centers specifically involved in treating PSM with regards to OC with peritoneal metastates. Despite encouraging results showing better short- as well as long-term outcomes of CRS combined with HIPEC as compared to the traditional management of advanced OC, some limitations are moved to data interpretation due to the enrollment of patients at multiple clinical scenarios and at progressive stages of the disease [147,174].

The retrospective Italian multicenter study tried to fill that gap by the analysis of data for consecutive patients with advanced ovarian cancer submitted to CRS plus HIPEC between 1998 and 2014 at different treatment time points, such as primary debulking surgery, after partial/complete and no response to neoadjuvant treatments, with the exclusion of cases with recurrence [175]. Overall, out of the 511 enrolled patients from 11 Italian PSM centers, 226 (44.2%) were treated for primary advanced cancer. In 70.8% of cases, peritonectomy procedures were able to carry out complete cytoreduction, with the lowest complete cytoreduction rate in patients undergoing interval debulking surgery and no responders to neoadjuvant therapy.

After a mean follow up period of 53.8 months, the Kaplan–Meier survival analysis for overall groups (excluding the cases treated for recurrence) indicated 5-year OS of 48.5% and PFS of 25.3%, significantly higher when compared with the results obtained from patients treated for recurrence. For patients treated for primary advanced OC with peritoneal metastates and interval CRS after neoadjuvant therapy, both combined with HIPEC, a same prognosis was registered (OS 57.2 vs. 43.3%, median 61.2 vs. 53.2 months). Interestingly, patients who even partly responded to neoadjuvant therapy had a significantly higher OS than those who did not respond (OS 47.6 vs. 24.5%, respectively) [176,177,178]. Additionally, in line with the available literature, patients with a pathological complete response to neoadjuvant therapy registered encouraging survival outcomes after CRS and HIPEC, highlighting the prognostic role of the neoadjuvant therapy response and open to defining a more tailored criteria for preoperative therapy indications (Figure 4b).

## 5. Upfront Debulking Surgery versus Interval Debulking Surgery for Advanced Tubo-Ovarian High-Grade Serous Carcinoma and Diffuse Peritoneal Metastases Treated with CRS and HIPEC

Several sources of bias such as low recruitment per years of participating centers, low rates of complete resection, and low expected survival outcomes, negatively affected the results of the EORTC [71] and CHORUS trial [72], thereby impeding the extension of encouraging results of neoadjuvant chemotherapy followed by interval CRS rather than primary CRS to the general clinical setting of patients with primary advanced OC [162,179]. Such a literature need poses the basis for a Italian multicentric retrospective study aimed at investigating the treatment outcomes in patients with advanced tubo-ovarian high-grade serous cancer and diffuse peritoneal metastases at diagnosis, undergoing CRS with peritonectomy procedures in primary or interval debulking settings combined with HIPEC [180]. Overall, among the 144 enrolled patients from the six Italian PSM centers, 34 (23.6%) underwent primary CRS and 110 patients (76.4%) underwent interval CRS, both combined with HIPEC. In the neoadjuvant group, restaging, according to the RECIST criteria [143], showed that 66.4% of patients had partial responses, 16.3% had stable disease and 17.3% had a complete clinical response. Interestingly, while in 70.1% of the whole cohort treatment required high-complexity surgical procedures (CRS), 84.2% of the patients who had a complete clinical response to neoadjuvant chemotherapy required intermediate complexity procedures, highlighting an inversely related effect of medical therapy responses to the number of peritonectomy procedures needed and its greater surgical-related morbidity as well. At a median 66.3-month follow-up, patients who underwent primary or interval debulking had similar outcomes. However, the survival analysis stratifying for neoadjuvant responses showed that non-responders had a significantly greater risk of recurrence than those who underwent primary CRS (HR 2.1217, 95% CI, 1.0885–4.136; *p*-value 0.027) and a higher risk of death (HR, 2.4525, 95% CI, 1.1219–5.361, *p*-value 0.025). A troubling result was the high recurrence rate (42.5%) of peritoneal disease in overall patients despite HIPEC, with the evidence that patients who underwent interval CRS had lower recurrence rates than those who responded partly or poorly to neoadjuvant chemotherapy (Figure 4c).

## 6. Comparison of Treatment Protocols Including Six vs. Three Cycles of Neoadjuvant Chemotherapy Followed by CRS and HIPEC in Primary High-Grade Serous Ovarian Cancer with Peritoneal Metastases (OVANAC–HIPEC)

The OVANAC–HIPEC study is an ongoing SICO multicenter retrospective study. Patients with primary, FIGO IIIC/IVA, high-grade serous OC, undergoing CRS+HIPEC between 2005 and 2018, are categorized according to the neoadjuvant chemotherapy plan (three vs. six cycles). One of the main criticisms to the OVHIPEC−1 trial was the high rate of bowel resection and protective stoma performed. Preliminary data from SICO Oncoteam Centers indicate that performing six cycles of neoadjuvant chemotherapy may maximize the effects of preoperative chemotherapy and reduce the needs of visceral resections. Furthermore, this may be associated with potential benefits in the long-term. This is also the rationale of the previously cited CHRONO trial; anyway, in such trial HIPEC treatment is not included in the design. Post-operative morbidity, recurrence rates and patterns and overall survival will be compared in the two groups under study. Furthermore, a propensity score-matched analysis in a subgroup of enrolled patients will compare such endpoints when adjusting for preoperative data such as age, BMI, CA125 level, presence of ascites, PCI and clinical nodal status (Figure 5a). This analysis may be necessary, because poor responders to short neoadjuvant therapy might be excluded from CRS and HIPEC, while similar patients undergoing a long regimen could obtain a further response and, consequently, be submitted to surgery. This potential selection bias will be considered in data interpretation and stratified analysis.

## 7. CRS and HIPEC in Advanced Ovarian Cancer (CHORINE Study)

The CHORINE study will focus on the management of patients with stage IIIC unresectable OC with partial or complete response after neoadjuvant chemotherapy (three cycles carboplatin–paclitaxel). After CC0−1, patients will be randomized to receive HIPEC (cisplatin 100 mg/m^2^ of body surface area and paclitaxel 175 mg/m^2^ of body surface area for 90 min at 42 °C) or CRS alone. The outcome of this study will be published shortly [181] (Figure 5b).

## 8. Conclusions

The present review contextualizes the experience, evidence and ongoing trials conducted by the SICO PSM Oncoteam. In most cases, PSM are considered terminal diseases with limited therapeutic options and poor prognosis. In highly specialized treatment centers, combined CRS and HIPEC demonstrate a positive emerging trend towards encouraging survival rates in patients with advanced ovarian cancer. CRS and HIPEC represent an aggressive multidisciplinary approach, aimed at removing all visible macroscopic disease and treat microscopic residual tumors. However, the paucity of large randomized trials is at the origin of the controversies regarding its widespread adoption. Over the past few years, the SICO PSM Oncoteam worked steadily in an attempt to describe the optimal extent of surgical resection associated with HIPEC, the best method to perform HIPEC and planning the most appropriate sequence of treatment. Several questions still remain to be answered, such as indications, timing, type and dose of anticancer drugs to be used in HIPEC treatment and so on. This provides a good starting point for discussion and further research.

## Figures and Tables

**Figure 1 cancers-14-06010-f001:**
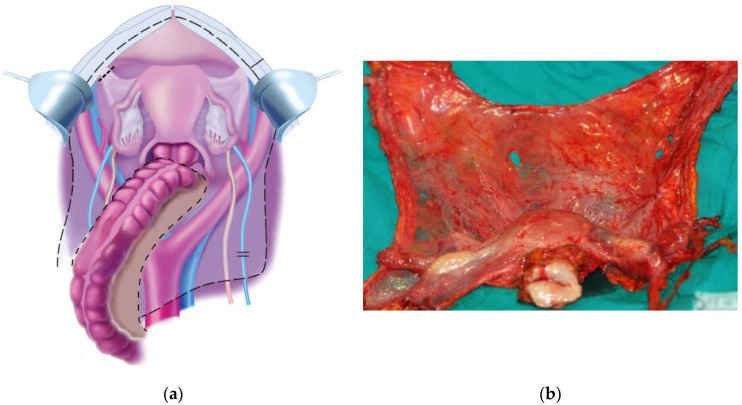
(**a**) Pelvic peritonectomy with en-bloc 7ysteron-adnexectomy. Dotted line represents the standard peritoneal resection margin after neoadjuvant chemotherapy with downstaging of peritoneal metastases; in most cases, no bowel resection is necessary to obtain a CCR-0 pelvic cytoreduction. Reproduced from Sugarbaker P. H. (2013). Adapted with permission from AME Publishing Company (accessed on 6 May 2022). (**b**) Pelvic peritoneum removed en-bloc with uterus and adnexa. Note the “bat shape” aspect of the resected specimen [128].

**Figure 2 cancers-14-06010-f002:**
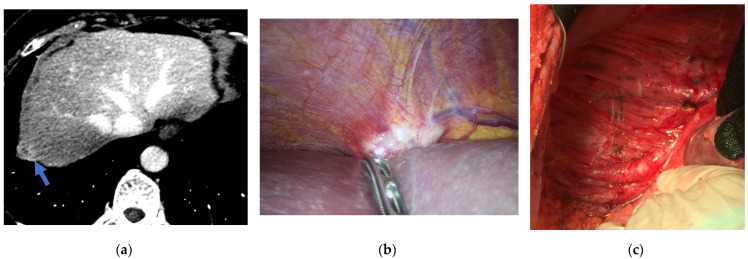
Female patient with peritoneal metastases from ovarian cancer: (**a**) CT of the abdomen showing a nodule of peritoneal metastases in right subphrenic region. (**b**) Laparoscopic PCI assessment confirming the imaging findings. (**c**) Completed right subphrenic peritonectomy.

**Figure 3 cancers-14-06010-f003:**
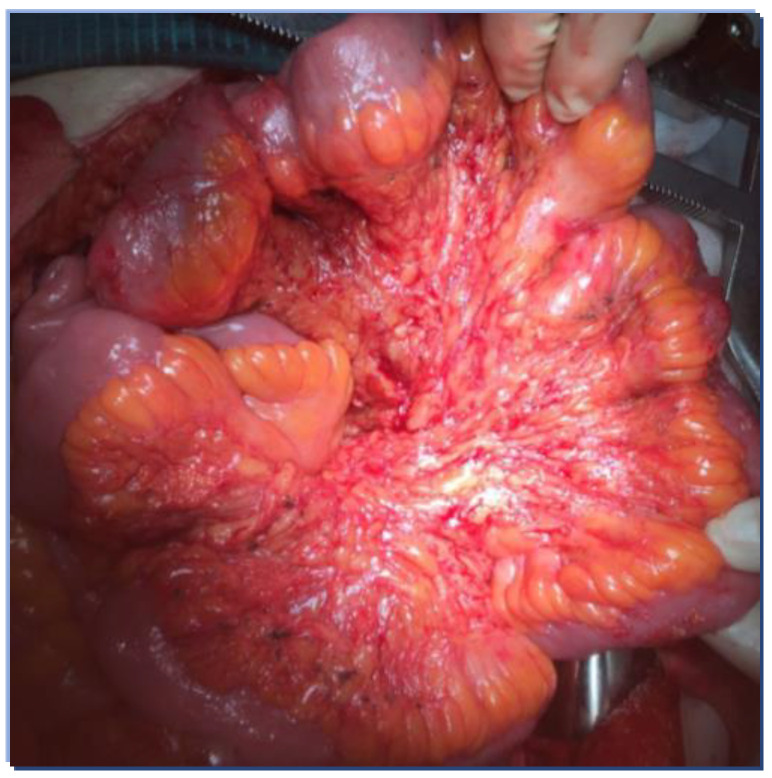
Completed mesenteric peritonectomy of the small bowel.

**Figure 4 cancers-14-06010-f004:**
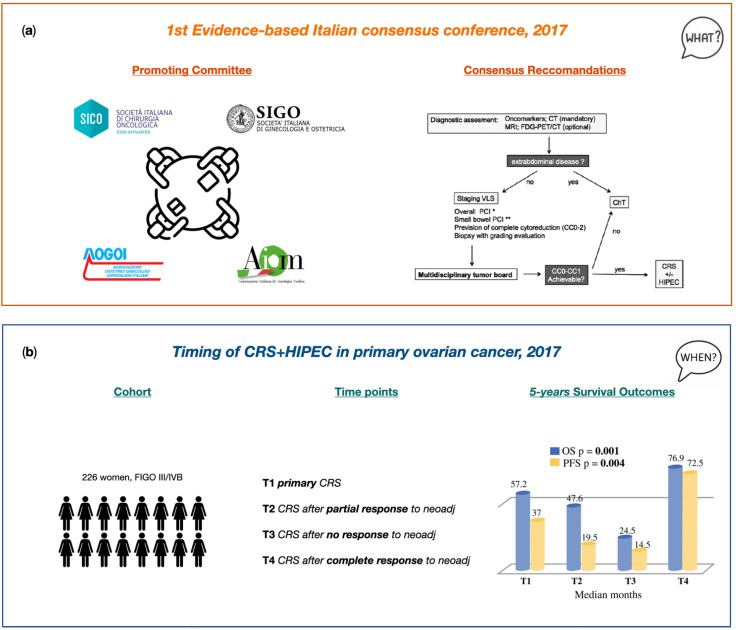
Italian PSM Oncoteam evidence. (**a**) Cavaliere et al. “1st Evidence−based Italian consensus conference on cytoreductive surgery and hyperthermic intraperitoneal chemotherapy for peritoneal carcinosis from ovarian cancer”. This image is licensed under the STM Guidelines http://www.stm-assoc.org/copyright-legal-affairs/permissions/permissions-guidelines/ (accessed on 16 May 2022). Reproduced with permission from Cavaliere D. (2017). (**b**) Di Giorgio et al. “Cytoreduction (Peritonectomy Procedures) Combined with Hyperthermic Intraperitoneal Chemotherapy (HIPEC) in Advanced Ovarian Cancer: Retrospective Italian Multicenter Observational Study of 511 Cases”. This image is licensed under the Creative Commons Attribution http://creativecommons.org/licenses/by/4.0 (accessed on 16 May 2022). Reproduced with permission from Di Giorgio A. (2016). (**c**) Biacchi et al. “Upfront debulking surgery versus interval debulking surgery for advanced tubo-ovarian high-grade serous carcinoma and diffuse peritoneal metastases treated with peritonectomy procedures plus HIPEC”. VLS: videolaparoscopy, ChT: chemotherapy, neoCht: neoadjuvant chemotherapy, PCI: peritoneal cancer index, CC: complete cytoreduction, FIGO: International Federation of Gynecology and Obstetrics, OS: overall survival, PFS: progression-free survival, CRS: cytoreductive surgery, HIPEC: hyperthermic intraperitoneal chemotherapy.

**Figure 5 cancers-14-06010-f005:**
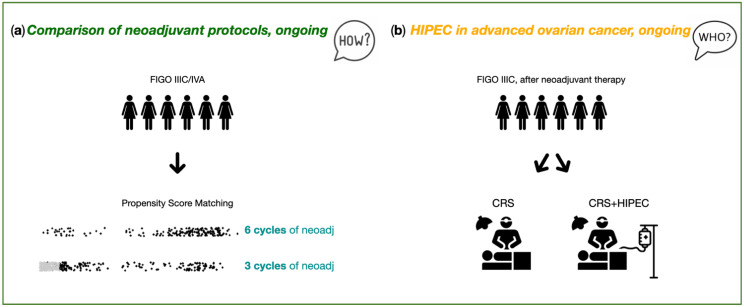
Italian PSM Oncoteam study purposes. (**a**) Marrelli et al. “OVANAC–HIPEC”. (**b**) Ansaloni et al. “CHORINE”. FIGO: International Federation of Gynecology and Obstetrics, neoadj: neoadjuvant therapy, CRS: cytoreductive surgery, HIPEC: hyperthermic intraperitoneal chemotherapy.

**Table 1 cancers-14-06010-t001:** FIGO staging classification for cancer of the ovary [21].

Stage		TNM
**I**	**Tumor confined to ovaries or fallopian tube(s).**	**T1 N0 M0**
IA	Tumor limited to one ovary (capsule intact) or fallopian tube; no tumor on ovarian or fallopian tube surface; no malignant cells in the ascites or peritoneal washings.	T1a N0 M0
IB	Tumor limited to both ovaries (capsules intact) or fallopian tubes; no tumor on ovarian or fallopian tube surface; no malignant cells in the ascites or peritoneal washings.	T1b N0 M0
IC	Tumor limited to one or both ovaries or fallopian tubes, with any of the following:	
*IC1*	*Surgical spill,*	*T1c1 N0 M0*
*IC2*	*Capsule ruptured before surgery or tumor on ovarian or fallopian tube surface,*	*T1c2 N0 M0*
*IC3*	*Malignant cells in the ascites or peritoneal washings.*	*T1c3 N0 M0*
**II**	**Tumor involves one or both ovaries or fallopian tubes with pelvic extension (below pelvic brim) or peritoneal cancer.**	**T2 N0 M0**
IIA	Extension and/or implants on uterus and/or fallopian tubes and/or ovaries.	T2a N0 M0
IIB	Extension to other pelvic intraperitoneal tissues.	T2b N0 M0
**III**	**Tumor involves one or both ovaries or fallopian tubes, or peritoneal cancer, with cytologically or histologically confirmed spread to the peritoneum outside the pelvis and/or metastasis to the retroperitoneal lymph nodes.**	**T1-3 N0-1 M0**
*IIIA1*	*Positive retroperitoneal lymph nodes only (cytologically or histologically proven):*	*T1-2 N1 M0*
*IIIA1(i)*	*Metastasis up to 10 mm in greatest dimension,*	
*IIIA1(i)*	*Metastasis more than 10 mm in greatest dimension,*	
*IIIA2*	*Microscopic extrapelvic (above the pelvic brim) peritoneal involvement with or without positive retroperitoneal lymph nodes.*	*T3a2 N0-1 M0*
IIIB	Macroscopic peritoneal metastasis beyond the pelvis up to 2 cm in greatest dimension, with or without metastasis to the retroperitoneal lymph nodes.	T3b N0-1 M0
IIIC	Macroscopic peritoneal metastasis beyond the pelvis more than 2 cm in greatest dimension, with or without metastasis to the retroperitoneal lymph nodes (includes extension of tumor to capsule of liver and spleen without parenchymal involvement of either organ).	T3c N0-1 M0
**IV**	**Distant metastasis excluding peritoneal metastases.**	**AnyT anyN M1**
IVA	Pleural effusion with positive cytology.	
IVB	Parenchymal metastases and metastases to extra-abdominal organs (including inguinal lymph nodes and lymph nodes outside of the abdominal cavity).	

**Table 2 cancers-14-06010-t002:** Completed trials. CRS: cytoreductive surgery; AC: adjuvant chemotherapy; NAC: neoadjuvant chemotherapy; HIPEC: hyperthermic intraperitoneal chemotherapy; OS: overall survival; PFS: progression-free survival.

Trial Code	Center	Treatment	Results
EORTC55971 [71]	Belgium	CRS + AC vs. NAC + CRS + AC in stage IIIC or IV	NAC followed by interval CRS is non-inferior to primary CRS followed by AC regarding OS and PFS.
CHORUS [72]	UK	CRS + 6 cycles of AC vs. 3 cycles of NAC + CRS + 3 cycles of AC in stage III or IV	Primary chemotherapy followed by interval CRS is non-inferior to primary CRS regarding OS.
JCOG0602 [74]	Japan	CRS + 8 cycles of AC vs. 4 cycles of NAC + CRS + 4 cycles of AC in stage III or IV	The non-inferiority of NAC was not confirmed.
SCORPION [75]	Italy	CRS + AC vs. NAC + CRS + AC in stage IIIC or IV	Primary CRS is non-inferior to NAC followed by interval CRS regarding OS and PFS, with different post-operative complications.
MSKCC [73]	USA	CRS + AC vs. NAC + CRS + AC in stage IIIC or IV	NAC should be reserved for patients who cannot tolerate primary CRS and/or for whom optimal CRS (≤1 cm residual) is not feasible.
CARCINOHIPEC [123]	Spain	NAC + CRS vs. NAC + CRS + HIPEC in FIGO IIIB/C	The addition of HIPEC to CRS improves OS and PFS.
OVHIPEC−1 [140]	Netherlands	NAC + CRS + AC vs. NAC + CRS + HIPEC + AC in stage III	HIPEC, combined with CRS, improves OS and PFS, and does not have higher rates of side effects.
Lim C.M. et al. [141]	South Korea	(NAC +) CRS vs. (NAC +) CRS + HIPEC in FIGO III or IV	The addition of HIPEC to CRS does not improve OS and PFS. Anyway, NAC subgroup has a trend of improved survival in favor of HIPEC.

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
