# Peer review of "Cytoreductive Surgery (CRS) and HIPEC for Advanced Ovarian Cancer with Peritoneal Metastases: Italian PSM Oncoteam Evidence and Study Purposes"

_cancers, 2022, doi:10.3390/cancers14236010_

Round 1

Reviewer 1 Report

This is a broad overview of advanced ovarian cancer management especially focusing on HIPEC from Italian SICO. This review summarized conclusively about the role of surgery and HIPEC in advanced ovarian cancer. Below please find some comments.

1. It will be better to include “HIPEC” in the title of this manuscript as the main focus is HIPEC.

2. Figure 1. The logo of SICO is unnecessary in this manuscript.

3. Line 134. “three cycles before plus six additional cycles” is not a fixed way of platinum-combined chemotherapy. “three to six additional cycles” may be better.

4. Page 357. The reason HIPEC has been shown to be valid in improving (3) quality of life, is not clear. Please clearly show the reference.

5. Figure 5. (c) corresponds to (a), (e) corresponds to (b).

Author Response

Dear Reviewer,
thank you very much for your pertinent suggestions.

Reviewer 2 Report

Marrelli and colleagues presented a very interesting review article describing the therapeutic results obtained by the Italian PSM Oncoteam in advanced ovarian cancer with peritoneal metastases. Overall, the authors described almost all the relevant aspects of ovarian cancer. The authors comprehensively described the therapeutic strategies available for metastatic OC by reporting the data of the main trials in this field. Overall, the manuscript is very interesting, below are reported some minor revisions that will further improve the quality of the paper:

1) In the Introduction or Discussion (Conclusions) sections, the authors have to better describe the therapeutic approaches available for OC by describing the novel PARP-inhibitors and other targeted therapies or monoclonal antibodies available (e.g. bevacizumab);

2) In the Introduction or Discussion (Conclusions) sections, the authors have to indicate how germline and somatic BRCA mutations influence the prognosis of patients and their response to therapies. At present almost all the germline mutations were identified and collected into databases and several studies demonstrated different metastatic rates and response rates in patients harboring pathogenic BRCA mutations. For this purpose, please see:

- PMID: 27403072

- PMID: 35383859

- PMID: 36063278

- PMID: 32131779

- PMID: 30909618

3) The authors should add a table reporting all the studies cited in the review article. The table should contain the Trial code, the center, the type of treatment, the main results obtained, the reference.

Author Response

Dear Reviewer,
thank you very much for your insightful remarks.
